# The Role of Ubiquitination in NF-κB Signaling during Virus Infection

**DOI:** 10.3390/v13020145

**Published:** 2021-01-20

**Authors:** Kun Song, Shitao Li

**Affiliations:** Department of Microbiology and Immunology, Tulane University, New Orleans, LA 70112, USA; ksong2@tulane.edu

**Keywords:** NF-κB, polyubiquitination, linear ubiquitination, inflammation, host defense, viral infection

## Abstract

The nuclear factor κB (NF-κB) family are the master transcription factors that control cell proliferation, apoptosis, the expression of interferons and proinflammatory factors, and viral infection. During viral infection, host innate immune system senses viral products, such as viral nucleic acids, to activate innate defense pathways, including the NF-κB signaling axis, thereby inhibiting viral infection. In these NF-κB signaling pathways, diverse types of ubiquitination have been shown to participate in different steps of the signal cascades. Recent advances find that viruses also modulate the ubiquitination in NF-κB signaling pathways to activate viral gene expression or inhibit host NF-κB activation and inflammation, thereby facilitating viral infection. Understanding the role of ubiquitination in NF-κB signaling during viral infection will advance our knowledge of regulatory mechanisms of NF-κB signaling and pave the avenue for potential antiviral therapeutics. Thus, here we systematically review the ubiquitination in NF-κB signaling, delineate how viruses modulate the NF-κB signaling via ubiquitination and discuss the potential future directions.

## 1. Introduction

The nuclear factor κB (NF-κB) is a small family of five transcription factors, including RelA (also known as p65), RelB, c-Rel, p50 and p52 [1]. All NF-κB transcription factors are featured with a conserved Rel homology domain (RHD) responsible for nuclear localization, DNA binding and dimerization. These transcription factors form a homodimer or heterodimer to bind to a specific DNA element, called κB sites. For example, p65 and p50 form a p65/p50 heterodimer, a major NF-κB complex in most cells. It has been reported that NF-κB regulates the expression of more than 400 human genes, including cytokines, chemokines, genes involved in stress response, cell growth, and apoptosis (see the website: https://www.bu.edu/nf-kb/gene-resources/target-genes/). Thus, the NF-κB signaling pathways profoundly impact many physiological and pathological processes, such as inflammation and host defense.

Ubiquitin is a small protein of 76 amino acids, highly conserved and ubiquitously expressed in eukaryotic cells. Ubiquitination is a reversible post-translational modification by adding one or more ubiquitin to the substrate protein [2]. Ubiquitination plays a diverse role in protein degradation, transcription, protein–protein interaction, signalosome scaffold, and subcellular localization. It is well-known that several types of ubiquitination tightly regulate NF-κB signaling pathways and are critical for the activation of NF-κB. As NF-κB is one of the master transcription factors controlling host defense, viruses have evolved strategies to modulate NF-κB signaling to either evade host surveillance or exploit it for viral gene expression. These emerging viral evasion strategies will help us better understand the virus–host interactions and provide potential antiviral targets. Thus, it is necessary to review recent advances and the insights from these studies for future directions. Here, we focus on several important polyubiquitination types and their roles in NF-κB signaling pathways and how viruses target these ubiquitinations to modulate NF-κB activity.

## 2. NF-κB Signaling Pathways

NF-κB is inactive in resting cells due to the cytosolic sequestration by the inhibitors of κBs (IκBs). The typical IκBs, such as IκBα and IκBβ, block the nuclear localization sequence (NLS) of NF-κB to retain it in the cytoplasm [3]. However, NF-κB is rapidly activated and translocated to the nucleus upon stimulations, such as TNF, viral infection, and UV. The activation of NF-κB is mainly through two distinct signaling branches: the canonical pathway and the noncanonical pathway (Figure 1) [1]. Both pathways are important for controlling viral infection and regulating immune and inflammatory responses. The canonical NF-κB activation is a downstream part of the signaling cascades of many signaling pathways elicited by diverse stimuli, including inflammatory cytokines and pathogen-associated molecular patterns (PAMPs). Although the upper part of these signaling cascades, such as receptors and adaptors, are diverse, they all activate the IκB kinase (IKK) complex consisting of two catalytic subunits, IKKα and IKKβ, and a regulatory subunit named NF-κB essential modulator (NEMO) (also known as IKKγ) [4]. The activated IKK complex further phosphorylates two serines of IκBα at the N-terminus, thereby triggering ubiquitin-dependent IκBα degradation in the proteasome and resulting in the release and translocation of NF-κB into the nucleus. The p50/p65 and p50/c-Rel dimers are the predominant NF-κB transcription factors inducing gene expression in the canonical pathway.

By contrast, the noncanonical NF-κB pathway is only activated by a narrow set of stimuli, such as the ligands of TNFR superfamily members lymphotoxin beta receptor (LT-βR), B-cell activating factor (BAFF), CD40, RANK, and the viral latent membrane protein 1 (LMP1) of Epstein–Barr virus (EBV). Furthermore, unlike the canonical pathway, IκBα degradation is not involved in the noncanonical NF-κB pathway. Instead, the noncanonical NF-κB pathway involves in the processing of the NF-κB2 precursor p100 to the mature form p52 [5]. Upon stimulation, the NF-κB-inducing kinase (NIK) activates IKKα, and then the activated IKKα phosphorylates p100, leading to its polyubiquitination and processing p52 by partial degradation of its C-terminal ankyrin repeats. The p52 forms a dimer with RelB and translocates into the nucleus to induce gene expression. Despite the different mechanisms in canonical and noncanonical NF-κB pathways, they both are regulated by ubiquitination [6,7].

## 3. Ubiquitination

### 3.1. Ubiquitination Process

The ubiquitination process is a three-step enzymatic reaction (Figure 2A). First, the E1 enzyme activates ubiquitin by linking the C-terminus of ubiquitin to E1 via a thioester bond in an ATP dependent manner. Second, the activated ubiquitin is transferred to the E2 conjugating enzyme, forming a thioester bond with the activated ubiquitin. Last, the E2 forms a complex with the E3 ligating enzyme, and the E2–E3 enzyme complex conjugates the ubiquitin to the substrate protein through an isopeptide bond between the lysine of the substrate protein and the C-terminal glycine of ubiquitin. Recently, it is found that cysteine, serine and threonine residues, as well as the free amino group of the N-terminus of proteins, also function as sites for ubiquitination through forming thioester, hydroxyester and peptide bonds, respectively [8]. In the human genome, there are two E1s (UBA1 and UBA6), about 35 E2s and more than 600 E3s [9]. The E3 ligases are diverse, and most of them consist of an interacting domain that determines the specificity of the substrate.

The E3 ligases are subdivided into three distinct classes: the really interesting new gene (RING), the homologous to the E6-associated protein C terminus (HECT), and the RING-between-RING (RBR) [10,11]. The RING E3 ligases are different from HECT and RBR ligases in two aspects. First, RING ligases bind the E2 and the substrate simultaneously and transfer the ubiquitin directly from the E2 to the substrate. By contrast, HECT and RBR ligases form an E3-ubiquitin intermediate via the active site cysteine in these E3 ligases. Then, the ubiquitin is transferred to a substrate of the E3 ligase. Second, in addition to functional monomers, RING ligases form homodimers and heterodimers and can be a part of large multisubunit complexes, such as the cullin-RING ligase (CRL) superfamily.

Ubiquitination is a reversible post-translational modification (Figure 2A). The removal of a ubiquitin or polyubiquitin chain from protein is mediated by deubiquitinases (DUBs), a group of proteases specifically targeting ubiquitin. The human genome has more than fifty DUBs, which are grouped into two main classes: the metalloproteases (JAB1/MPN/Mv34 metalloenzymes (JAMMs)) and the cysteine proteases [ubiquitin-specific protease (USP), ubiquitin C-terminal hydrolases (UCHs), MIU-containing novel DUB family (Mindy), Machado–Joseph disease proteases (MJDs), ovarian tumor proteases (OTUs)] [12]. Like E2 and E3, DUBs also display specificity toward one or a few linkages. For example, the OTU domain-containing deubiquitinase with linear linkage specificity (OTULIN) only hydrolyzes the linear polyubiquitin chain [13,14]. Overall, the E3 and DUBs provide an on–off switch controlling ubiquitination, enabling a regulatory mechanism by external and internal signals.

### 3.2. Ubiquitination Types

Diverse types of ubiquitination have been found in recent years [10,15] (Figure 2B). First, they can be grouped into mono-ubiquitination and polyubiquitination according to the amount of ubiquitin on the lysine residue of the protein substrate. Mono-ubiquitination is a single ubiquitin molecule conjugated to a lysine of the substrate, whereas polyubiquitination is the conjugation of a ubiquitin polymer to the substrate. Proteins also can be modified at multiple lysine residues with a single ubiquitin molecule, which is called multi-monoubiquitylation. Second, the polyubiquitin chains are grouped into different linkages determined by the conjugation through one of their lysine residues (K6, K11, K27, K29, K33, K48, K63) in the ubiquitin. In addition to these seven chain types, the N-terminal methionine residue (M1) of the ubiquitin can be conjugated to the carboxyl terminus of another ubiquitin forms a Met1-linked polyubiquitin chain (also known as linear polyubiquitination). Third, the polyubiquitin chains can also be classified into homotypic and heterotypic chains. The homotypic chains comprise only a single linkage type. By contrast, heterotypic chains contain mixed linkages within the same polymer. Last, one ubiquitin molecule can be ubiquitinated at two or more sites to form a branched polyubiquitin chain.

The type of ubiquitination determines the fate of the substrate, resulting in different outcomes. For example, the K11- and K48-linked polyubiquitin chains usually drive proteasomal degradation, whereas the monoubiquitylation and K63-linked polyubiquitination mainly involve in protein–protein interaction, DNA damage response, protein kinase activation, and the regulation of protein localization [10]. K27-linked polyubiquitination is involved in the DNA damage response and innate immunity, whereas K29-linked polyubiquitin inhibits Wnt signaling [15]. In addition, K33 linkages have been implicated in post-Golgi protein trafficking [16]. Several types of polyubiquitination tightly regulate NF-κB signaling pathways, including K48- and K63-linked polyubiquitination, Met1-linked polyubiquitination, and unanchored polyubiquitin, which are reviewed in detail below with a focus on the E3 ligases and DUBs involved in NF-κB pathways.

#### 3.2.1. K48-Linked Polyubiquitination

In most scenarios, K48-linked polyubiquitination leads to proteasomal degradation of the target protein. K48-linked polyubiquitination is the first characterized polyubiquitination and plays an essential role in both canonical and noncanonical NF-κB pathways. In the canonical NF-κB pathway, the critical step is the phosphorylation-induced ubiquitination and degradation of IκBs, allowing NF-κB dimers to translocate into the nucleus [1]. Phosphorylation of IκB at serines 32 and 36 creates a degron recognized by the E3 ubiquitin ligase complex comprising Skp, cullin, F-box (SCF) [17,18]. Mechanistically, phosphorylation of IκBs recruits the β-transducing repeat-containing protein (β-TrCP) that bridges an SCF complex consisting of S-phase kinase-associated protein 1 (Skp1), cullin, RING-box protein 1 (Rbx1) and the E2 conjugating enzymes UBCH5b, UBCH5c or UBCH3 to IκB. This complex catalyzes K48-linked polyubiquitination of IκB proteins, resulting in IκB degradation by 26S proteasome and the subsequent release of NF-κB dimers into the nucleus. The K48-linked ubiquitination of IκBs is also reversed by a couple of DUBs. USP11 and USP15 have been shown to deubiquitinate IκB in the TNF-induced NF-κB pathway [19,20]. However, it is not clear whether these DUBs also are involved in other canonical NF-κB pathways.

In the noncanonical NF-κB pathway, the processing of p100 for generating p52 is also dependent on K48-linked polyubiquitination [17,18]. Interestingly, p100 has a similar motif to that one found in IκB proteins. After the motif is phosphorylated by IKKα, p100 is also recognized by the SCF-β-TrCP E3 ligase complex, which leads to partial degradation of the C terminal ankyrin repeats of p100 by the proteasome to make p52. Recent studies further found that p100 can be fully degraded by K48-linked ubiquitination via different sites and E3 ligase. Regarding this complete degradation pathway, the p100 protein is first phosphorylated by glycogen synthase kinase-3 (GSK3). Then, these phosphorylated sites recruit another SCF E3 ligase complex, SCF-Fbw7 [21,22]. The SCF-Fbw7 induces K48-linked polyubiquitination and triggers the complete degradation of p100 by the proteasome.

K48-linked polyubiquitination is the most common type of polyubiquitination and also widely regulates the upstream of all NF-κB pathways, which is discussed in Section 4. In addition, unlike the K63-linked polyubiquitination and linear polyubiquitination discussed below, the E2 conjugating enzymes for K48-linked polyubiquitination are not limited to one or two specific enzymes. Similarly, the E3 ligases for K48-linked polyubiquitination are also diverse and have much less specificity on E2.

#### 3.2.2. K63-Linked Polyubiquitination

In NF-κB pathways, K63-linked polyubiquitin plays a critical role in stabilizing the receptor signalosome on the membrane, facilitating the recruitment of downstream adaptors or complexes, and activating kinases [7]. For example, in the IL-1-induced NF-κB signaling, TRAF6 functions as an E3 ligase to catalyze the synthesis of K63-linked polyubiquitin with the E2 enzyme Ubc13-Uev1A, leading to the activation of the TAK1 kinase complex, which in turn phosphorylates and activates IKK. Many E3 ligases in various NF-κB pathways are capable of catalyzing the K63-linked polyubiquitin; however, it was first thought that Ubc13-Uev1A is the sole E2 enzyme specifically for K63-linked polyubiquitination. Furthermore, genetic studies found that NF-κB activation mediated by TNFα, IL-1β, and TLR ligands is almost normal in several types of cells isolated from Ubc13 knockout mice [23]. Of note, another study showed that IL-1β-induced IKK activation is reduced in Ubc13 knockout mouse embryonic fibroblasts (MEFs) [24]. To reconcile the discrepancy, a study further showed that IL-1-induced NF-κB activation was dependent on Ubc13 while TNFα-induced NF-κB activation was dependent on Ubc5, an E2 enzyme that can synthesize heterogeneous polyubiquitin chains [25].

#### 3.2.3. Met1-Linked Polyubiquitination

Met1-linked or linear polyubiquitination is a new type of ubiquitin modification in NF-kB pathways [26]. Several proteins in NF-κB pathways are the known targets for Met1-linked polyubiquitination, including NEMO, RIP1, FADD, and BCL10. The Met1-linked polyubiquitin chain is assembled by the linear ubiquitin chain assembly complex (LUBAC), a ubiquitin E3 ligase complex comprising HOIL-1 interacting protein (HOIP), heme-oxidized IRP2 ubiquitin ligase 1 (HOIL-1), and SHANK-interacting protein-like 1 (SHARPIN) [27,28,29,30,31] together with the E2 conjugating enzyme UBE2L3 [32]. LUBAC is the only known E3 ligase that catalyzes Met1-linked polyubiquitin; however, LUBAC also mediates K48-linked ubiquitination of TRIM25, an E3 ligase for the cytosolic RNA sensor, retinoic acid-inducible gene I (RIG-I) [33].

#### 3.2.4. Unanchored Polyubiquitin

Studies showed that the unanchored polyubiquitin chains, which are not conjugated to any other cellular proteins, can activate TGFβ-activated kinase 1 (TAK1) in vitro. The unanchored K63-linked polyubiquitin chains directly activate the TAK1 kinase complex in vitro through binding to the TAK1-binding protein 2 (TAB2) or TAB3 subunit of this complex. The activated TAK1 then phosphorylates IKKβ, leading to IKK activation [34]. Similarly, K63-linked polyubiquitin also activates RIG-I in an in vitro reconstitution system [35]. Nonetheless, more work is needed to elucidate protein kinase activation mechanism by unanchored polyubiquitin chains and their roles in vivo.

## 4. Ubiquitination in NF-κB Signaling Pathways

As mentioned above, many stimuli activate NF-κB pathways and the signal cascades upstream of IKK and NIK are diverse and complex, although they are also regulated by ubiquitination. Thus, this review only focuses on two examples, TNFR1 and RIG-I signaling pathways (Figure 3). TNFR1 is a well-studied inflammatory cytokines-induced signaling pathway, while RIG-I is the major antiviral innate immune pathway in response to RNA virus infection. These pathways are not only critical for inflammation and host defense during viral infection but also are extensively examined for polyubiquitin-mediated regulation.

### 4.1. Ubiquitination in the TNFR1 Signaling Pathway

The TNF receptor 1 (TNFR1)-signaling pathway is the most well-studied canonical NF-κB pathway. TNFR1 is a receptor for the inflammatory cytokine, tumor necrosis factor-alpha (TNFα), which induces inflammation, apoptosis, and necrosis. Once engaged with TNFα, TNFR1 trimerizes and recruits different cellular adaptors to form complex I and complex II, leading to NF-κB activation and cell death, respectively. To fit the review topic, we focus on the signaling pathway elicited by TNFR1 complex I. Within the complex I, the tumor necrosis factor receptor type 1-associated death domain protein (TRADD) binds TNFR1 through the death domain. TRADD further recruits the receptor-interacting serine/threonine–protein kinase 1 (RIPK1) and the E3 ligases TNF receptor-associated factor 2 (TRAF2), TRAF5, and c-IAPs. cIAPs or TRAFs generate K63-linked polyubiquitin chains on TRAF2 and RIP1. The K63-linked polyubiquitin chains then recruit the TAK1-TAB complex and the IKK complex to the receptor complexes via K63-selective binding of TAB2/3 or NEMO, respectively. Recent studies also showed that the LUBAC E3 ligase complex is also recruited to the TNFR1 and generates Met1-linked polyubiquitin on RIP1 and NEMO [26]. These Met1-linked polyubiquitin chains also function as a scaffold to recruit the IKK complex via the ubiquitin-binding domain of NEMO. Within the TNFR1 signalosome complex, TAK1 activates the IKK by phosphorylation, and then the activated IKK complex phosphorylates IκBα. The phosphorylated IκBα is ubiquitinated by the E3 complex SCFβ-TrCP for the K48-ubiquitination-mediated proteasomal degradation of IκBα. After liberation from IκBα, the canonical NF-κB transcription factors, predominantly composed of homo- or hetero-dimers of p65 (RelA) and p50, translocate into the nucleus and activate NF-κB target genes (Figure 3).

DUBs are also engaged with a opposite role to E3 ligases by balancing ubiquitination modification. The K63-linked polyubiquitin is hydrolyzed by the cylindromatosis gene CYLD and A20 (also known as TNFα-induced protein 3). CYLD contains a USP domain in the C-terminus, which mediates the cleavage of polyubiquitins. In the NF-κB pathways, CYLD removes the K63-linked polyubiquitin chains from TAK1, RIP1, BCL3, TRAF2, TRAF6, and NEMO. Unlike CYLD, A20 is a hybrid of DUB and E3 ligase, which has an N-terminal OTU domain responsible for polyubiquitin cleavage and a C-terminal domain of seven Cys2-Cys2 zinc-fingers (ZF) that render the E3 ligase activity. A20 disassembles K63-linked polyubiquitin chains of RIP1, TRAF6, RIPK2, NEMO and MALT1, thus inhibiting the activation of NF-κB activation. A20 also mediates K48-linked ubiquitination of RIP1, leading to its degradation, thus inhibits NF-κB activation. The cellular zinc finger anti-NF-κB (Cezanne) is a deubiquitinase of the ovarian tumor superfamily with sequence similarity to A20. Cezanne also suppresses NF-κB activation by targeting RIP1 signaling intermediaries for deubiquitination [36].

In addition, the ubiquitin-specific peptidase 21 (USP21) inhibits NF-κB in the TNF pathway by deubiquitination of RIP1 [37]. USP31 and USP7 also inhibit NF-κB signaling by deubiquitinating K63-linked polyubiquitin chains [38,39]. MCP-induced protein 1 (MCPIP1) (also known as Zc3h12a) deubiquitinates TRAF proteins and negatively regulates JNK and NF-κB signaling [40]. USP11 and USP15 remove K48-linked chains from IκBα, thus protecting IκBα from degradation by the proteasome [19,20]. Interestingly, USP2 positively regulates TNF-induced NF-κB activation; however, the target of USP2 is unknown [41].

The Met1-linked polyubiquitin chain is cleaved by CYLD and OTULIN. CYLD exhibits deubiquitinase activity toward multiple types of polyubiquitin; however, OTULIN specifically targets Met1-linked polyubiquitin. OTULIN also interacts with HOIP via its PUB-interacting motif and inhibits LUBAC activity [42,43]. Recently, we found that TRIM32 conjugates nonproteolytic polyubiquitin onto OTULIN and the polyubiquitin blocks the interaction between HOIP and OTULIN, thereby activating NF-κB signaling [44].

### 4.2. Ubiquitination in the RIG-I Signaling Pathway

The innate immune system uses pattern recognition receptors (PRRs) in different cellular compartments to sense microbial components that mark invading viruses. Many of these PRRs have been characterized, including the RIG-I-like receptors (RLRs), such as RIG-I, MDA5 and LGP2 [45,46]. The RLRs recognize viral RNA in the cytoplasm, for example, the double-stranded RNA (dsRNA) or 5’ triphosphate RNA generated by viral replication of RNA viruses [47,48,49,50,51,52]. The engagement of viral RNA induces the conformational change of RIG-I and several post-translational modifications, which leads to the oligomerization of RIG-I. The oligomerized RIG-I translocates to the mitochondria and binds the mitochondrial antiviral signaling protein (MAVS, also known as CARDIF, IPS1, and VISA). The binding results in the oligomerization of MAVS and the recruitment of TANK-binding kinase 1 (TBK1). The oligomerized MAVS further recruits TBK1, TRAF6, IKK and interferon regulatory factor 3 (IRF3). Subsequently, TBK1 phosphorylates IRF3, which in turn triggers its dimerization and nuclear translocation [53,54,55,56,57], whereas IKK phosphorylates IκBα to release NF-κB. In the nucleus, IRF3 and NF-κB, together with other transcription factors, form active transcriptional complexes and activate type I IFN expression [53,54,55,56,58] (Figure 3).

Similar to the TNFR1 signaling pathway, the RIG-I signaling pathway is also heavily regulated by ubiquitination. First, RIG-I undergoes several types of ubiquitination, which is regulated by at least nine E3 ligases and four DUBs. Three ubiquitin E3 ligases, TRIM25 [59], MEX3C [60], and TRIM4 [61], have been reported to activate RIG-I signaling by mediating K63-linked polyubiquitination of the N-terminal CARD domain of RIG-I. TRIM25 was first found to bind the CARD and mediate polyubiquitination of CARD at Lys172 [59]. Other studies showed that MEX3C and TRIM4 target RIG-I for ubiquitination at different sites [60,61]. Recently, a study showed that another ubiquitin E3 ligase, RIPLET (a.k.a. REUL), is the predominant E3 ligase for RIG-I K63-linked ubiquitination and activation [62,63,64]; however, RIPLET ubiquitinates the CTD domain of RIG-I at Lys849 and Lys851 [65]. Interestingly, unanchored K63-linked polyubiquitin is also found to non-covalently bind the CARD and promote CARD tetramerization and concomitant signal activation [66]. These activating K63-linked polyubiquitin chains of RIG-I can be removed by several DUBs, including USP3, USP21 and CYLD [67,68,69,70]. RIG-I also can be conjugated with K48-linked polyubiquitin. The E3 ligases RNF122, RNF125, c-Cbl, and CHIP have been found to mediate K48-linked ubiquitination and proteasomal degradation of RIG-I, thereby inhibiting RIG-I signaling [71,72,73,74]. Another E3 ligase TRIM40 conjugates both K27- and K48-linked polyubiquitin onto RIG-I for proteasomal degradation [75]. The K48-linked ubiquitination of RIG-I is reversed by USP4 [76]. In addition, LUBAC mediates K48-linked ubiquitination of TRIM25, an E3 ligase for RIG-I K63-linked ubiquitination and activation. The K48-linked ubiquitination leads to TRIM25 protein degradation, thereby inhibiting RIG-I activation [33]. In contrast, USP15 deubiquitylates TRIM25, preventing the LUBAC-dependent degradation of TRIM25 [77].

The downstream of RIG-I is also regulated by ubiquitination. As with RIG-I, MAVS is also ubiquitinated multiple E3 ligases, although most of them mediate K48-linked ubiquitination and proteasomal degradation. Two HEC domain-containing E3 ligases, the Smad ubiquitin regulatory factor 1 (Smurf1) and Smurf2, also mediate K48-linked ubiquitination of MAVS [78,79]. Other E3 ligases, including MARCH5, RNF5, AIP4, TRIM25, and pVHL, are all reported to promote MAVS protein degradation through K48-linked polyubiquitination [80,81,82,83]. Unlike the redundancy of E3 ligases for K48-linked polyubiquitination, there is only one E3 ligase TRIM31 was reported to confer the K63-linked polyubiquitin onto MAVS and promotes MAVS aggregation and activation [84]. TRIM21 promotes innate immune response to RNA viral infection through K27-linked polyubiquitination of MAVS [85]. There are two DUBs for MAVS, the ovarian tumor family deubiquitinase 4 (OTUD4) and YOD1, cleaves K48- and K63-linked polyubiquitin chains of MAVS, respectively [86,87]. Interestingly, TRIM44, an unusual DUB, stabilizes MAVS by preventing its ubiquitination and degradation [88]. In addition, K63-linked ubiquitination of TRIM14 bridges NEMO to MAVS [89].

Interestingly, two E3 ligases catalyze non-K48-linked polyubiquitin chains on MAVS to promote its autophagic degradation. The E3 ubiquitin ligase MARCH8 catalyzes the K27-linked polyubiquitin chain on MAVS at lysine 7, which serves as a recognition signal for NDP52-dependent autophagic degradation [90]. Another E3 ligase RNF34 initiates the K63- to K27-linked polyubiquitin switch on MAVS at Lys 311, thus facilitating the autophagic degradation of MAVS [91].

TBK1 is the key kinase for the IRF3 signaling axis of the RIG-I pathway. We and others showed that TBK1 is K63-linked ubiquitinated for activation by E3 ligases mind bomb 1 (MIB1), MIB2, and RNF128 [92,93]. Overall, the RIG-I signaling pathway is heavily regulated by different types of ubiquitination, which provides multiple layers of regulation for this pathway.

## 5. Modulation of the Ubiquitination in NF-κB Pathways by Viruses

### 5.1. Virus-Mediated Inhibition of NF-κB Signaling by Modulation of Ubiquitination

#### 5.1.1. Virus-Encoded E3 Ligases and DUBs

Since ubiquitination plays an essential role in the NF-κB signaling pathways, viruses have evolved diverse strategies that target the ubiquitination to inhibit the NF-κB signaling pathways to evade the host immune surveillance (Figure 3). First, some virus genomes encode a viral E3 ligase that directly targets the core components of NF-κB pathways for proteasomal degradation, thus shutting off NF-κB signaling. For example, the ICP0 of herpes simplex virus 1 (HSV-1) is an E3 ligase with a RING finger domain at its N-terminus [94]. ICP0 interacts with p50 and targets it for K48-linked ubiquitination, leading to suppression of the NF-κB activity [95]. ICP0 also promotes the degradation of MyD88 and TRIAP, leading to the inhibition of Toll-like receptor 2 (TLR2)-mediated NF-κB pathway [96]. The nonstructural protein 1 (NSP1) of rotavirus is also a putative E3 ligase and has been shown to induce the ubiquitination-dependent proteasomal degradation of SCF-β-TrCP, thus stabilizing IκB and inhibiting NF-κB activation [97,98].

Second, some other viruses also encode a viral DUB to directly remove the K63-linked polyubiquitin chain to block NF-κB involved signaling pathways. The papain-like protease (PLP) of severe acute respiratory syndrome coronavirus (SARS-CoV) inhibits IRF3 and NF-κB activation by removing the K63-linked polyubiquitin chain of TRAF3 and TRAF6 [99,100,101]. The leader proteinase (Lpro) from foot-and-mouth disease virus (FMDV) has a similar topology and DUB activity with the PLP of SARS-CoV. Lpro inhibits type I IFN production by deubiquitinating the K63-linked polyubiquitin of RIG-I, TBK1, TRAF3 and TRAF6 [102]. The ORF64 of Kaposi’s sarcoma-associated herpesvirus (KSHV) cleaves the K63-linked polyubiquitin of RIG-I and counteracts the host’s innate immune defense [103]. Similarly, the ORF64 of murine gamma herpesvirus 68 (MHV68) also encodes a DUB and antagonizes the cGAS-STING signaling pathway, although the target proteins are not clear [104]. Moreover, HCMV UL48 interacts with RIP1 and removes polyubiquitin chains from RIP1 [105].

Virus-encoded DUBs also cleaves the K48-linked polyubiquitin chain to subvert NF-κB activation. The NSP2 of PRRSV contains an OUT domain and disrupts the K48-linked ubiquitination of IκBα, thus preventing IκBα degradation and inhibiting NF-κB activation [106,107]. The UL36 of HSV-1 also removes polyubiquitin chains from IκBα [108]. Interestingly, the BPLF1 of EBV cleaves both K48- and K63-linked polyubiquitin chains. Specifically, BPLF1 chops K48-linked polyubiquitin from IκBα and K63-linked polyubiquitin from TRAF6 and NEMO [109].

#### 5.1.2. Viruses Hijack Host Proteolytic Ubiquitination

Viruses also hijack the host ubiquitination system. First, viruses hijack host E3 ligases. For example, poxvirus protein MC132 bridges the host elongin-B/elongin-C/cullin-5 ubiquitin ligase complex to p65, leading to p65 ubiquitination and degradation [110]. Furthermore, the murid herpesvirus-4 ORF73 protein and KSHV latency-associated nuclear antigen 1 also cause polyubiquitination and degradation of p65 by interacting with the host elongin-B/elongin-C/cullin-5 ubiquitin ligase complex, inhibiting its binding to kB sequences [111,112]. The ORF-9b of SARS-CoV targets the MAVS signalosome by usurping the E3 ligase AIP4 to trigger the degradation of MAVS, TRAF3, and TRAF6 [113]. Interestingly, the ICP0 of HSV-1 interacts with USP7 to prevent ICP0 autoubiquitination and protein degradation, which leads to an efficient HSV-1 lytic infection [114].

In addition, the X protein of hepatitis B virus (HBV) binds to MAVS and promotes its degradation through the ubiquitination of Lys 136 [115]. Furthermore, the NS1 and NS2 proteins of respiratory syncytial virus (RSV) assemble a heterogeneous degradative complex to trigger the proteasome-dependent degradation of RIG-I and other immune molecules [116].

#### 5.1.3. Viruses Subverts Host K63-Linked and Met1-Linked Ubiquitination

Viruses can inhibit the K63- and Met1-linked ubiquitination by several distinct mechanisms. First, viruses can target host E3 ligases responsible for K63-linked ubiquitination in NF-κB signaling pathways. For example, the NS3–NS4A protease complex of HCV cleaves Riplet to prevent the K63-ubiquitylation of RIG-I [117].

Second, viral proteins also can interact with host E3 ligases and perturb their E3 ligase activity. For example, the NS1 proteins of the influenza A virus interacts with TRIM25 and Riplet and inhibits TRIM25- and Riplet-mediated K63-linked polyubiquitination of RIG-I [118,119]. The murine cytomegalovirus (MCMV) protein M45 interacts with the DNA-dependent activator of the IRFs (DAI) and RIP1 and disrupts the ubiquitination of RIP1, leading to the blocking of NF-kB activation [120,121]. The HBV polymerase interacts with STING and disrupts the K63-linked ubiquitination of STING to block the DNA-sensing pathway [122]. HBV e antigen binds NEMO and interferes with the TRAF6-dependent K63-linked polyubiquitination of NEMO [123]. Similarly, the nonstructural protein 3 (NS3) of the hepatitis C virus (HCV) competes for the binding of NEMO to LUBAC, leading to the decreased Met1-linked polyubiquitination of NEMO and NF-κB activation [124].

Third, viral proteins can block the interaction between polyubiquitin and ubiquitin-binding proteins, thus blocking the downstream signaling. For example, the MC005 protein of molluscum contagiosum virus (MCV) binds NEMO and blocks polyubiquitin chains binding to NEMO, thereby inhibiting IKKβ activation [125].

Fourth, viral proteins modulate the expression of host E3 ligases and DUBs. For example, the open reading frame 3 (ORF3) of hepatitis E virus (HEV) reduces the expression of TRAF6 to suppress the activation of the NF-κB pathway [126]. The HEV ORF3 protein also decreases the expression of TRADD protein and K63-linked polyubiquitination of RIP1 [127]. By contrast, the measles virus P protein can increase the expression of the DUB A20. The accumulated A20, in turn, removes polyubiquitin chains from ATF6, thus blocking the activation of NF-κB signaling [128].

Last, viral RNAs also participate in the regulation of host ubiquitination of NF-κB pathways. For example, the 3C protein of enterovirus 71 (EV71) downregulates the host microRNA miR-526a, leading to an increased expression of the DUB enzyme CYLD and subsequent inhibition of the activation of RIG-I [129].

### 5.2. Virus-Mediated Activation of NF-κB Signaling by Modulation of Ubiquitination

During evolution, some viruses also have developed strategies to activate the NF-kB signaling pathway for viral replication or cell proliferation (Figure 3). For example, KSHV infection has been shown to induce the activation of the NF-kB signaling pathway. The expression of viral GPCR induces TAK1 phosphorylation and K63-linked polyubiquitination, which activates the NF-kB signaling pathway and promotes the KSHV malignancies [130]. The viral protein R (Vpr) of HIV-1 interacts with TAK1. This interaction promotes the polyubiquitination of TAK1, leading to its enzymatic activation [131]. The saimiri transforming protein C (STP-C) of *Herpesvirus saimiri* interacts with TRAF6 protein and induces polyubiquitination of TRAF6, leading to the NF-kB activation and IL-8 production, which creates an inflammatory microenvironment for chronic activation [132]. The LMP1 of EBV induces an association between TRAF1 and the linear ubiquitin chain assembly complex (LUBAC) [133]. The interaction stimulates Met1-linked polyubiquitin chain attachment to TRAF1 complexes, which promotes NF-κB activation and is critical for EBV-mediated B-cell transformation [133]. Human papillomavirus E6 induces protein degradation of NFX1-91, which binds to the promoter of the NF-κB inhibitor p105 and upregulates its expression. Thus, the E6-mediated NFX1-91 degradation activates NF-κB [134]. The Tax protein of human T-cell lymphotropic virus type 1 (HTLV-1) activates NF-κB via both K63- and Met1-linked ubiquitination [135,136].

## 6. Conclusions and Perspectives

In response to infection, inflammation and innate immunity are the major strategies used by the host to limit viral infection. NF-κB is one of the major transcription factors that control inflammatory and innate immune responses to fight against invasion by upregulating the expression of chemokines, cytokines, adhesion molecules and enzymes that produce secondary inflammatory mediators. Various types of ubiquitination regulate NF-κB activation and play a critical role in host defense to viral infection. Interestingly, as summarized above, viruses adopt distinct evasion strategies by targeting ubiquitination. Although viral modulation of K48- and K63-linked ubiquitination has been found in many viruses, targeting Met1-linked ubiquitination by viruses is just emerging. It will also be interesting to investigate whether viruses interfere with the unconjugated free polyubiquitin, a new form of ubiquitination critical for RIG-I and IKK activation.

Ubiquitination process is reversibly controlled by two opposing enzyme groups, E3 ligases and DUBs, which makes ubiquitination an ideal drug target. To limit viral infection, one approach is to develop small molecules that specifically target viral E3 ligases and DUBs. Other approaches can target the substrate-binding interface and modulate the gene expression of E3 ligases and DUBs. NF-κB is essential for host defense; however, over-activation of NF-κB pathways causes inflammatory and autoimmune diseases. A few inhibitors targeting the E3 ligases in NF-κB pathways have been developed. For example, the LUBAC inhibitors, HOIPINs, suppressed Met1-linked polyubiquitination and reduced the expression of proinflammatory factors [137]. Nonetheless, with the growing knowledge of the role and mechanisms of ubiquitination in NF-κB pathways and viral infection, more potent and specific drugs targeting either viral or host E3 ligases and DUBs will be developed for infectious and inflammatory diseases.

## Figures and Tables

**Figure 1 viruses-13-00145-f001:**
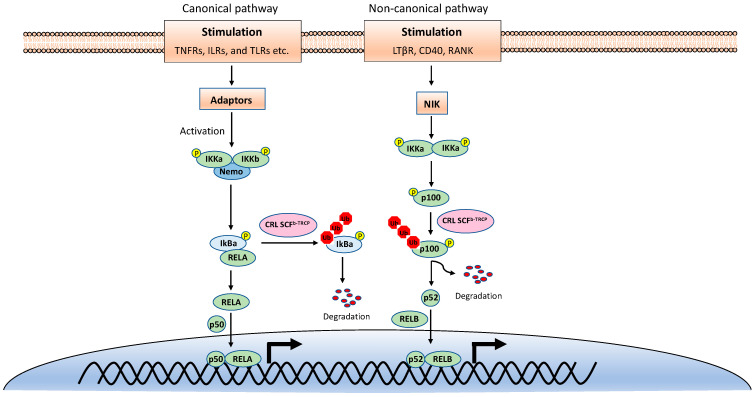
The canonical and noncanonical nuclear factor κB (NF-κB) pathways.

**Figure 2 viruses-13-00145-f002:**
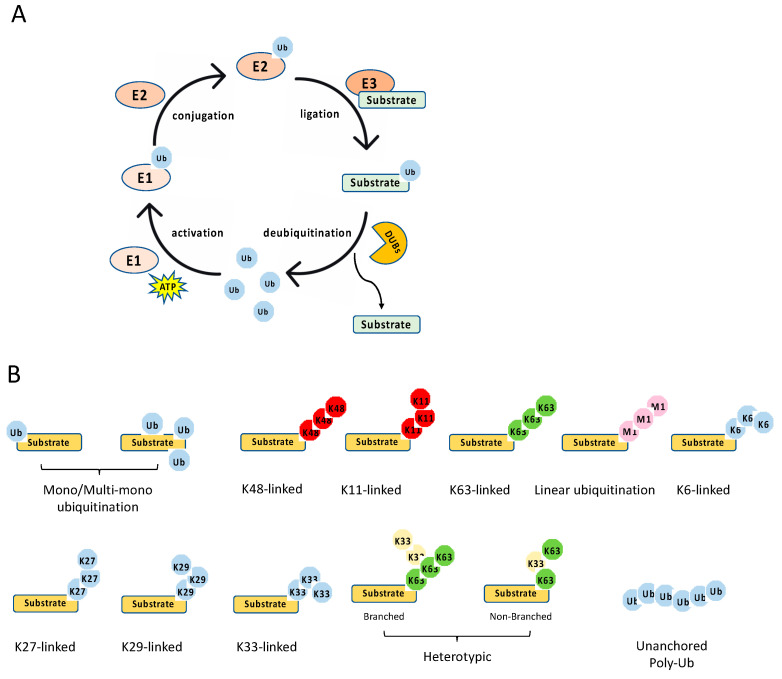
Ubiquitination process and types. (**A**) The multiple steps of the enzymatic process for ubiquitination controlled by E1, E2, E3, and deubiquitinases (DUBs). (**B**) The schematics of various types of ubiquitination.

**Figure 3 viruses-13-00145-f003:**
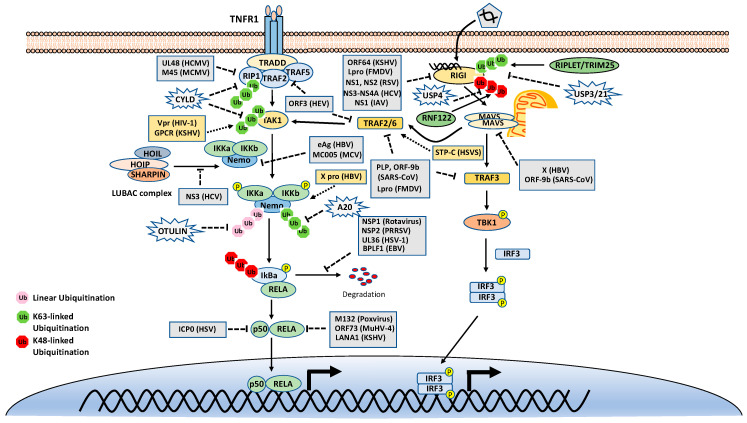
Tumor necrosis factor receptor type 1 (TNFR1) and retinoic acid-inducible gene I (RIG-I) signaling pathways. K48-, K63-, and Met1-linked polyubiquitinations are illustrated. Virus proteins that suppress or activate NF-κB via modulation of ubiquitination are indicated.

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
