# Peer review of "The Role of Ubiquitination in NF-κB Signaling during Virus Infection"

_viruses, 2021, doi:10.3390/v13020145_

Round 1

Reviewer 1 Report

This review article by Song and Li focuses on the role of ubiquitination in NF-κB signaling, and how viruses modulate the NF-κB signaling via ubiquitination. It is built upon key and up-to-date literature that describes ubiquitination in TNFR1 and RIG-I signaling pathways and modulation of the ubiquitination in NF-κB pathways by viruses.  The authors explained different types of ubiquitination, mono-ubiquitination and polyubiquitination, including the most important K48-linked polyubiquitination, K63-linked polyubiquitination, Met1-linked polyubiquitination and unanchored polyubiquitin. Finally, the authors point out the importance in study on NF-kB signaling pathway ubiquitination in inflammation and innate immunity responses to viral infection, and the urgency of gaining our knowledge on Met1-linked ubiquitination and unanchored polyubiquitin by viruses.

Overall, the review is nicely written and largely reflects our current understanding and open questions in the field. Here are, however, a number of issues that should be addressed to improve the manuscript:

  1. The title is somewhat misleading. The authors may change it to “The role of ubiquitination in NF-κB signaling during or against virus infection”.

  1. Figure 2B is not clear. It is recommended that the authors re-draw the figure using the same color for the types of ubiquitination and their corresponding outcomes. For example, the K11- and K48-linked polyubiquitin chains usually drive proteasomal degradation. Use green icons represent both K48-linked polyubiquitin and proteasomal degradation.

  1. Line 142-145. There are two “however” in the two sentences. This should be rephrased.

  1. Line 105-128 mentioned the K48-linked polyubiquitination in both canonical and non-canonical NF-κB pathways. However, the canonical and non-canonical NF-κB pathways didn’t get introduced until Section 3.1(Line 168-194). The authors should consider moving Section 3.1 ahead of Section 2.1.

  1. It would be more informative that the important E3 ligases, such as TRIM25 and RIPLET, are added to Figure 3. The figure should be re-made to contribute with all knowledge provided in the review.

  1. Line 395-398. “NF-kB initiates inflammatory and innate immune responses to fight against invasion by upregulating the production of chemokines, cytokines, adhesion molecules and enzymes that produce secondary inflammatory mediators.” The statement is not true. It is well established that pattern recognition receptors (PRRs) recognize viral invasion and initiate innate immune responses, including inflammatory responses. NF-kB pathways are downstream signaling of PRRs (RIG-I) or cytokines (TNF).This part of summary needs to be modified.

  1. Line 43. Add references after “…ubiqutination in NF-κB signaling pathways to facilitate viral infection.” Also, add references in Line 210-239.

  1. It would be interesting to know if there are ubiquitination patterns in NF-κB signaling pathways among different innate immune cells types, such as epithelial cells, dendritic cells and macrophages. The authors could highlight more recent findings in this aspect.

Author Response

We thank the reviewer for critical reading and constructive suggestions. Our point by point reply is as follows:

  1. The title is somewhat misleading. The authors may change it to “The role of ubiquitination in NF-κB signaling during or against virus infection”.

We made the change in the title.

  1. Figure 2B is not clear. It is recommended that the authors re-draw the figure using the same color for the types of ubiquitination and their corresponding outcomes. For example, the K11- and K48-linked polyubiquitin chains usually drive proteasomal degradation. Use green icons represent both K48-linked polyubiquitin and proteasomal degradation.

 We think the reviewer referred it to Fig. 1B. We changed K11 and K48 linked polyubiquitin to red color, and K63 linked polyubiquitin to green color.

  1. Line 142-145. There are two “however” in the two sentences. This should be rephrased.

 We changed one “however” to “furthermore”.

  1. Line 105-128 mentioned the K48-linked polyubiquitination in both canonical and non-canonical NF-κB pathways. However, the canonical and non-canonical NF-κB pathways didn’t get introduced until Section 3.1(Line 168-194). The authors should consider moving Section 3.1 ahead of Section 2.1.

We changed the structure and moved section 3.1 (now it is section 2) before Section 2.1 (now it is 3.1).

  1. It would be more informative that the important E3 ligases, such as TRIM25 and RIPLET, are added to Figure 3. The figure should be re-made to contribute with all knowledge provided in the review.

We added TRIM25 and RIPLET in Fig. 3.

  1. Line 395-398. “NF-kB initiates inflammatory and innate immune responses to fight against invasion by upregulating the production of chemokines, cytokines, adhesion molecules, and enzymes that produce secondary inflammatory mediators.” The statement is not true. It is well established that pattern recognition receptors (PRRs) recognize viral invasion and initiate innate immune responses, including inflammatory responses. NF-kB pathways are downstream signaling of PRRs (RIG-I) or cytokines (TNF). This part of the summary needs to be modified.

We rephrased the sentence.

  1. Line 43. Add references after “…ubiquitination in NF-κB signaling pathways to facilitate viral infection.” Also, add references in Line 210-239.

 We deleted “…ubiquitination in NF-κB signaling pathways to facilitate viral infection.” References are added in Line 210-239.

  1. It would be interesting to know if there are ubiquitination patterns in NF-κB signaling pathways among different innate immune cells types, such as epithelial cells, dendritic cells and macrophages. The authors could highlight more recent findings in this aspect

This might be beyond the topic of this review.

Reviewer 2 Report

The Nf-kB signalling pathway is one of the major signalling transduction cascades that regulate cell proliferation and defences. Thus, viruses are targeting this cascade, often via targeting ubiquitination events. While recent years have seen a lot of research, a current and comprehensive review of viral manipulation of Nf-kB signalling is absolutely interesting.

However, in its current state the manuscript is sometimes hard to follow and jumps back and forth between subtopics. A clearer subdivision into accessible paragraphs would be helpful, too. Just as a suggestion I would restructure like this:

  • The current structure, that Ubiquitination is introduced first could be kept, except that all the specific parts regarding functions in the Nf-kB could be moved to the Nf-kB part. E.g. lines 107-117.
  • I personally would have expected a bit more information on the different types of ubiquitinations and why they differ in paragraph 2 and less focus on K48, K63 and for some reason Met1. It is not entirely clear, why besides K48 and K63 as the apparent choices Met1 was singled out and highlighted. In general, it would have been nicer, if all types of ubiquitination would have been covered and not just a few selected, as each type can be easily described in a few sentences.
  • In Paragraph 3, the signalling could be introduced briefly as multiple stimulatory pathways which converge in canonical or non-canonical signalling pathways according to the stimulus. Then a sub-paragraph could provide an overview over canonical signalling, another on non-canonical and a third one cover the stimuli with a brief introduction and the two mentioned examples.
  • The structure of Paragraph 4 is no immediately apparent. In fact, I am not really getting the structure at all and the subheadings are not seemingly logically connected. Again, K63 and Met1 linkages are highlighted in a separate paragraph. Could sorting by virus class makes sense? Section 4.2. sounds a bit like an introductory section for paragraph 4.

The overall lack of references is sometimes a bit confusing, as there are paragraphs which completely lack even a citation for a previous review, just as an example the whole first part of  section 3.2. is devoid of any reference to previous literature.

In Fig. 1B it is not entirely clear to what the circular text boxes are associated to, are they supposed to describe the functions of the ubiquitin above them or below them or both? The red boxes could be changed to not overlap with the sequence and the lysines/methionine e.g. highlighted in bold or via arrows.

Fig.2 ‘non-canonical’ signalling involved ubiquitin as well, please visualise in the Figure.

Minor comments:

Line 13 “on the contrary”: That phrase may sound a bit misleading here, as there are no obvious contrary statements given.

Line 66 “from”: is it meant as ‘of’, or ‘by’?

Line 107 “as discussed above”: This was not discussed above?

Line 136: I am not entirely sure what is meant by that sentence.

Line 172 “translocate”: translocated

Line 240: The linkage is referring to what?

Line 240-245: Could be omitted as the paragraph is a bit hard to understand and does not seem to provide any additional information required for this review.

Line 248 “discriminate among”: Why should those receptors discriminate among microbes and not rather sense microbials but not sense the host? I am not entirely clear what is meant by the current sentence.

Line 250: please insert a ‘for example’ when describing ligands for RLRs

Line 255-257: There is a sentence duplication, providing information on TBK1 twice.

Line 263: RIG-I

Line 285: The reference to what is ubiquitinated is missing in that sentence (MAVS, RIG-I?)

Lines 299-302: That section could be removed as it is not really relevant in the context of this review, or alternatively abbreviated to one sentence.

Line 314: The remark about IFI16 is not really required in the context of an Nf-kB focused review.

Paragraph 2.2.: In Ubiquitination types unanchored chains are not listed, though they are mention alter on in 2.2.4 as an ubiquitination type.

Paragraph 2.2: Chains or ubiquitin not conjugated to lysines are not covered and could be briefly mentioned.

In general, the same colour schemes as used to depict different ubiquitin linkage types could b e used throughout the figures. E.g. Fig. 3 vs Fig. 1

Author Response

We thank the reviewer for critical reading and constructive suggestions. Our point by point reply is as follows:

  • The current structure, that Ubiquitination is introduced first could be kept, except that all the specific parts regarding functions in the Nf-kB could be moved to the Nf-kB part. E.g. lines 107-117.

We re-structure the manuscript according to the first reviewer’s suggestion.

  • I personally would have expected a bit more information on the different types of ubiquitinations and why they differ in paragraph 2 and less focus on K48, K63 and for some reason Met1. It is not entirely clear, why besides K48 and K63 as the apparent choices Met1 was singled out and highlighted. In general, it would have been nicer, if all types of ubiquitination would have been covered and not just a few selected, as each type can be easily described in a few sentences.

We add a little bit more information on other linkages.

In Paragraph 3, the signalling could be introduced briefly as multiple stimulatory pathways which converge in canonical or non-canonical signalling pathways according to the stimulus. Then a sub-paragraph could provide an overview over canonical signalling, another on non-canonical and a third one cover the stimuli with a brief introduction and the two mentioned examples

We re-structure the manuscript according to first reviewer’s suggestion.

The structure of Paragraph 4 is no immediately apparent. In fact, I am not really getting the structure at all and the subheadings are not seemingly logically connected. Again, K63 and Met1 linkages are highlighted in a separate paragraph. Could sorting by virus class makes sense? Section 4.2. sounds a bit like an introductory section for paragraph 4.

Again, we highlight K48, K63 and Met1 because they are critical for NF-kB activation, which is evidenced by many genetic models. Other linkages might only play a minor role in NF-kB pathways. We add more description of other linkages, but it is important to deliver the cirtical information that not all linkages are equally important in NF-kB pathways.

The overall lack of references is sometimes a bit confusing, as there are paragraphs which completely lack even a citation for a previous review, just as an example the whole first part of  section 3.2. is devoid of any reference to previous literature.

Reference added

In Fig. 1B it is not entirely clear to what the circular text boxes are associated to, are they supposed to describe the functions of the ubiquitin above them or below them or both? The red boxes could be changed to not overlap with the sequence and the lysines/methionine e.g. highlighted in bold or via arrows.

The circular text boxes and red boxes are deleted in Fig. 1.

Fig.2 ‘non-canonical’ signalling involved ubiquitin as well, please visualise in the Figure.

We now add the ubiquitination process in non-canonical signaling in Fig. 2.

Minor comments:

Line 13 “on the contrary”: That phrase may sound a bit misleading here, as there are no obvious contrary statements given.

Deleted.

Line 66 “from”: is it meant as ‘of’, or ‘by’?

It is changed to “of”.

Line 107 “as discussed above”: This was not discussed above?

Deleted.

Line 136: I am not entirely sure what is meant by that sentence.

This sentence is deleted.

Line 172 “translocate”: translocated

Changed.

Line 240: The linkage is referring to what?

Met-1 linked ubiquitination is referring to linear ubiquitination.

Line 240-245: Could be omitted as the paragraph is a bit hard to understand and does not seem to provide any additional information required for this review.

This paragraph discussed the regulation of linear ubiquitination which is critical for TNFR induced NF-kB signaling.

Line 248 “discriminate among”: Why should those receptors discriminate among microbes and not rather sense microbials but not sense the host? I am not entirely clear what is meant by the current sentence.

We change the ‘discriminate among’ to ‘sense’.

Line 250: please insert a ‘for example’ when describing ligands for RLRs

Add ‘for example’ and rephrase the sentence.

Line 255-257: There is a sentence duplication, providing information on TBK1 twice.

Changed.

Line 263: RIG-I

Changed.

Line 285: The reference to what is ubiquitinated is missing in that sentence (MAVS, RIG-I?)

The sentence is deleted due to incorrect citation.

Lines 299-302: That section could be removed as it is not really relevant in the context of this review, or alternatively abbreviated to one sentence.

This section discussed the regulation of TBK1 which is an essential part of downstream of RIG-I signaling pathway by ubiquitination.

Line 314: The remark about IFI16 is not really required in the context of an Nf-kB focused review.

Deleted.

Paragraph 2.2.: In Ubiquitination types unanchored chains are not listed, though they are mention alter on in 2.2.4 as an ubiquitination type.

Added in Fig. 2B.

Paragraph 2.2: Chains or ubiquitin not conjugated to lysines are not covered and could be briefly mentioned.

Added.

In general, the same colour schemes as used to depict different ubiquitin linkage types could b e used throughout the figures. E.g. Fig. 3 vs Fig. 1

We now make the same color for the same linkage throughout all figures.

Reviewer 3 Report

This review focuses on the ubiquitination regulation of NFκB activation by TNFR1 and RIG-I pathways. It is a good review, and is well organized. The review should provide useful information in this field.

I have following main suggestions to improve the review:

  1. Since TNFR1 activation of NFκB is a very advanced topic, including ubiquitination regulation of this pathway that has been studied for decades. So please focus on the recent advances on this pathway. In this regard, Section 3.2.1, second paragraph needs to be intensively updated with more recent findings. The authors can detail the classes of DUBs first, and then show which of them are involved in this setting. For example, in humans, there are 15 OTU DUBs, which can be divided into three groups: OTUs, OTUBs, and A20-like OTUs. In the present version, only CYLD, A20, and OTULIN are shown in this section. CYLD and A20 are very well known in this setting. There are some new DUBs that have been identified to modify ubiquitination in this setting, including other members in the OTU family in addition to OTULIN and A20.
  2. E3 ligases and DUBs are substrate specific, and they are the key information in this review, rather than the TNFR1 and RIG-I signaling pathways themselves, which are well known already and appeared in an overwhelming number of publications. So, the mentioned E3 ligases and DUBs in this review are better be displayed in the figures.
  3. Section 2.2. Ubiquitination types. Recently, ubiquitination on non-lysine sites have been reported, and should be included here. The old types, K48 and K63, as long as the TNFR1 signaling pathway itself, however, I think they can be introduced more briefly in that they can be found in numerous reviews. In this regard, Fig 1 and Fig 2 can also be modified by adding new updates on these figures.
  4. In section 4, the authors try to expand viral regulation of ubiquitination-mediated NFκB activation in other contexts, by adding KSHV and EBV, two oncogenic viruses that activate NFκB in their latent infections. The authors need to clarify this point, i.e. latent vs lytic infection. Also, why do they not include HTLV1 that also activates NFκB in its latency? and probably so do other viruses? If the authors want to continue focusing only on TNFR1 and RIG-I in this section, it may be better to remove KSHV and EBV contents, and modify the subtitle to clarify this point.
  5. Lines 262-282. RIG-I ubiquitination. REUL and USP4 promote RIG-I activation in different ways, and TRIM38 stabilizes RIG-I and MDA5 by sumoylation. In addition, TRIM13 and TRIM59 were suggested to promote RIG-I activity. Any others? The authors need to search recent literatures to update.
  6. Lines 290-291. MAVS ubiquitination. As such, many other E3 ligases promote MAVS activation or stabilization, for example, many TRIM family members. TRIM14 undergoes K63-linked self ubiquitination, consequently recruiting NEMO to MAVS signalosome for NFκB activation. TRIM21/Ro52 triggers K27-linked ubiquitination, and TRIM31 triggers K63-linked ubiquitination, of MAVS, and TRIM44 stabilizes MAVS by preventing its ubiquitination and degradation, to promote RIG-I signaling pathway. Intensive updates are needed here.

Other minor issues:

  1. As mentioned above, it is for sure that the review cannot include viral modulation of ubiquitination in NFκB activation downstream of all pathways in all contexts, such as cGAS-STING. So section 4 needs to be more clear.
  2. Line 176. “The canonical NFκB pathway” to “The canonical NFκB activation”
  3. Line 187. Better to include examples of viral factors, for example, EBV LMP1 and HTLV1 Tax (LMP1 is a member of the TNFR superfamily but Tax may not), both activate canonical and non-canonical NFκB. In contrast to TNFR superfamily members which require only IKKα, Tax requires both IKKα and NEMO to activate the noncanonical NF-κB pathway via formation of a complex that also contains p100.
  4. Line 230. E3 ligases and DUBs are both engaged with opposite roles to balance ubiquitination modification, not only in TNFR1 signaling, but also in any ubiquitination-mediated process. Need more precise description here.
  5. Line 337. “Beside virus-encoded E3 ligases and DUBs, viruses also hijack host ubiquitination system”. Needs to be more precise. Some Viral E3 and DUBs are encoded for the purpose to hijack host ubiquitination system. For example, HSV1 ICP0 hijacks host USP7, a key DUB in the host ubiquitination system.
  6. Lines 387-391. The reference for the EBV case is missing.

Author Response

We thank the reviewer for critical reading and constructive suggestions. Our point by point reply is as follows:

  1. Since TNFR1 activation of NFκB is a very advanced topic, including ubiquitination regulation of this pathway that has been studied for decades. So please focus on the recent advances on this pathway. In this regard, Section 3.2.1, second paragraph needs to be intensively updated with more recent findings. The authors can detail the classes of DUBs first, and then show which of them are involved in this setting. For example, in humans, there are 15 OTU DUBs, which can be divided into three groups: OTUs, OTUBs, and A20-like OTUs. In the present version, only CYLD, A20, and OTULIN are shown in this section. CYLD and A20 are very well known in this setting. There are some new DUBs that have been identified to modify ubiquitination in this setting, including other members in the OTU family in addition to OTULIN and A20.

We added new OTUs.

  1. E3 ligases and DUBs are substrate specific, and they are the key information in this review, rather than the TNFR1 and RIG-I signaling pathways themselves, which are well known already and appeared in an overwhelming number of publications. So, the mentioned E3 ligases and DUBs in this review are better be displayed in the figures.

E3 ligase and DUBs are added to the figures.

  1. Section 2.2. Ubiquitination types. Recently, ubiquitination on non-lysine sites have been reported, and should be included here. The old types, K48 and K63, as long as the TNFR1 signaling pathway itself, however, I think they can be introduced more briefly in that they can be found in numerous reviews. In this regard, Fig 1 and Fig 2 can also be modified by adding new updates on these figures.

Non-lysine ubiquitination is added in the text.

  1. In section 4, the authors try to expand viral regulation of ubiquitination-mediated NFκB activation in other contexts, by adding KSHV and EBV, two oncogenic viruses that activate NFκB in their latent infections. The authors need to clarify this point, i.e. latent vs lytic infection. Also, why do they not include HTLV1 that also activates NFκB in its latency? and probably so do other viruses? If the authors want to continue focusing only on TNFR1 and RIG-I in this section, it may be better to remove KSHV and EBV contents, and modify the subtitle to clarify this point.

There are two subsections: one is virus-mediated inhibition of NF-kB signaling and the other is virus-mediated activation of NF-kB signaling We change the subtitles to clarify the confusion.

HTLV1 is added.

  1. Lines 262-282. RIG-I ubiquitination. REUL and USP4 promote RIG-I activation in different ways, and TRIM38 stabilizes RIG-I and MDA5 by sumoylation. In addition, TRIM13 and TRIM59 were suggested to promote RIG-I activity. Any others? The authors need to search recent literatures to update.

REUL is another name of RIPLET. These E3 ligases and DUB are added. Sumoylation is not the topic of this review. TRIM13 and TRIM59 are reported to negatively regulate MDA5; however, whether ubiquitination is involved is not clear in the paper.

  1. Lines 290-291. MAVS ubiquitination. As such, many other E3 ligases promote MAVS activation or stabilization, for example, many TRIM family members. TRIM14 undergoes K63-linked self ubiquitination, consequently recruiting NEMO to MAVS signalosome for NFκB activation. TRIM21/Ro52 triggers K27-linked ubiquitination, and TRIM31 triggers K63-linked ubiquitination, of MAVS, and TRIM44 stabilizes MAVS by preventing its ubiquitination and degradation, to promote RIG-I signaling pathway. Intensive updates are needed here.

These E3 ligases are added.

Other minor issues:

  1. As mentioned above, it is for sure that the review cannot include viral modulation of ubiquitination in NFκB activation downstream of all pathways in all contexts, such as cGAS-STING. So section 4 needs to be more clear.

We divide the section into two parts: one is viral modulation of ubiquitination in NFκB inhibition and the other is viral modulation of ubiquitination in NFκB activation.

  1. Line 176. “The canonical NFκB pathway” to “The canonical NFκB activation”

Changed.

  1. Line 187. Better to include examples of viral factors, for example, EBV LMP1 and HTLV1 Tax (LMP1 is a member of the TNFR superfamily but Tax may not), both activate canonical and non-canonical NFκB. In contrast to TNFR superfamily members which require only IKKα, Tax requires both IKKα and NEMO to activate the noncanonical NF-κB pathway via formation of a complex that also contains p100.

LMP1 is added.

  1. Line 230. E3 ligases and DUBs are both engaged with opposite roles to balance ubiquitination modification, not only in TNFR1 signaling, but also in any ubiquitination-mediated process. Need more precise description here.

Clarified.

  1. Line 337. “Beside virus-encoded E3 ligases and DUBs, viruses also hijack host ubiquitination system”. Needs to be more precise. Some Viral E3 and DUBs are encoded for the purpose to hijack host ubiquitination system. For example, HSV1 ICP0 hijacks host USP7, a key DUB in the host ubiquitination system.

The sentence is rephrased and the reference is added.

  1. Lines 387-391. The reference for the EBV case is missing.

 Added.

Reviewer 4 Report

In “The role of ubiquitination in NF-κB signaling and virus infection”, Song and Li review the scientific literature documenting the role ubiquitination in NF-κB signaling and how viruses use ubiquitination to modulate NF-κB responses. The topic is an interesting one that fits well with the theme of the Special Issue. However, there are some issues with the manuscript. Below are some comments to help improve it.

  1. Some of the references cited are other reviews. It would be better to cite original literature as that is what readers are seeking when they read a review article.
  2. The text in the figures is too small.
  3. Some references need to be added to certain sections. For example, lines 77, 98, 223
  4. Since the review focuses only on TNFR1 and RIG-I signaling, the abstract should mention that as well.
  5. The paper would benefit from a summary figure for sections 3.2.1 and 3.2.2 showing which cellular E3 ligase and DUBs affect TNFR1 and RIG-I signaling.
  6. There are several grammatical and sentence structure issues that need to be addressed so that the manuscript is easier to read.

Author Response

We thank the reviewer for critical reading and constructive suggestions. Our point by point reply is as follows:

  1. Some of the references cited are other reviews. It would be better to cite original literature as that is what readers are seeking when they read a review article.

Original literatures are cited now.

  1. The text in the figures is too small.

 We make it bigger now.

  1. Some references need to be added to certain sections. For example, lines 77, 98, 223

Added.

  1. Since the review focuses only on TNFR1 and RIG-I signaling, the abstract should mention that as well.

It might make the review off focus.

  1. The paper would benefit from a summary figure for sections 3.2.1 and 3.2.2 showing which cellular E3 ligase and DUBs affect TNFR1 and RIG-I signaling.

We add cellular E3 ligase and DUBs to the figure.

  1. There are several grammatical and sentence structure issues that need to be addressed so that the manuscript is easier to read.

We asked our colleagues to polish the manuscript.